# Blue Electroluminescence in SRO-HFCVD Films

**DOI:** 10.3390/nano11040943

**Published:** 2021-04-08

**Authors:** Haydee P. Martínez, José A. Luna, Roberto Morales, José F. Casco, José A. D. Hernández, Adan Luna, Zaira J. Hernández, Gabriel Mendoza, Karim Monfil, Raquel Ramírez, Jesús Carrillo, Javier Flores

**Affiliations:** 1Departamento de Ingeniería Eléctrica y Electrónica, Tecnológico Nacional de México/Instituto Tecnológico de Apizaco Carretera Apizaco-Tzompantepec, Esquina con Av. Instituto Tecnológico S/N. Conurbado Apizaco-Tzompantepec, Apizaco 90300, Mexico; haydee.mh@apizaco.tecnm.mx (H.P.M.); roberto.mc@apizaco.tecnm.mx (R.M.); federico.cv@apizaco.tecnm.mx (J.F.C.); 2Centro de Investigación en Dispositivos Semiconductores (CIDS-ICUAP), Benemérita Universidad Autónoma de Puebla (BUAP), Av. San Claudio y 14 sur, Edif. IC5 C.U., Col. San Manuel, Puebla 72570, Mexico; jose.hernandez@correo.buap.mx (J.A.D.H.); imezaira@gmail.com (Z.J.H.); gaomec13@gmail.com (G.M.); karim.monfil@correo.buap.mx (K.M.); jesus.jecarril@gmail.com (J.C.); 3Facultad de Ingeniería Química, Benemérita Universidad Autónoma de Puebla (BUAP), Puebla 72570, Mexico; lunaf86@gmail.com; 4Carrera de Mecatrónica, Universidad Tecnológica de Huejotzingo (UTH), Real San Mateo 36B, Segunda Secc, Santa Ana Xalmimilulco, Puebla 74169, Mexico; newraq77@hotmail.com; 5Departamento de Ingeniería, Benemérita Universidad Autónoma de Puebla-Ciudad Universitaria, Blvd. Valsequillo y Esquina, Av. San Claudio s/n, Col. San Manuel, Puebla 72570, Mexico; xavier_snk@hotmail.com

**Keywords:** electroluminescence, HFCVD, conduction mechanisms

## Abstract

In this work, electroluminescence in Metal-Insulator-Semiconductors (MIS) and Metal-Insulator-Metal (MIM)-type structures was studied. These structures were fabricated with single- and double-layer silicon-rich-oxide (SRO) films by means of Hot Filament Chemical Vapor Deposition (HFCVD), gold and indium tin oxide (ITO) were used on silicon and quartz substrates as a back and front contact, respectively. The thickness, refractive indices, and excess silicon of the SRO films were analyzed. The behavior of the MIS and MIM-type structures and the effects of the pristine current-voltage (I-V) curves with high and low conduction states are presented. The structures exhibit different conduction mechanisms as the Ohmic, Poole–Frenkel, Fowler–Nordheim, and Hopping that contribute to carrier transport in the SRO films. These conduction mechanisms are related to the electroluminescence spectra obtained from the MIS and MIM-like structures with SRO films. The electroluminescence present in these structures has shown bright dots in the low current of 36 uA with a voltage of −20 V to −50 V. However, when applied voltages greater than −67 V with 270 uA, a full area with uniform blue light emission is shown.

## 1. Introduction

Although in the field of optoelectronics there has been substantial research regarding electroluminescent Metal-Insulator-Metal (MIM) structures [1,2], the use of silicon-based materials, structures, and devices is a crucial advantage for optoelectronic applications due to the high-performance, and low-cost manufacturing techniques that are used [3,4,5,6], besides that silicon is the second most abundant chemical element in the earth. Silicon-based structures focused on optoelectronics include Si nanocrystals, Si nanowires, Ge alloys, and other methods to achieve the required carrier confinement to increase the efficiency of radiative recombination [7,8], and it is essential to mention that significant advances are also being realized with silicon oxide materials containing optically active rare-earth impurities [7,8].

Some of the most exciting applications of Si-nanostructures include optical emitters for integrated optical circuits, memory logic circuits; massively parallel optical interconnects and cross-connections for integrated circuit chips; light-wave components; high power matrix and discrete emitters; nano cell optoelectronic matrices to detect biological and chemical agents [9,10,11,12,13,14,15].

The semiconductor nanostructures based on silicon that absorb and emit light, such as porous silicon (PS), silicon nanowires, silicon oxide nitride (SiON), and silicon-rich oxide (SRO), present a great opportunity of being applied in electro photonics and they are compatible with Metal Oxide Semiconductors (MOS) technology [7,8,9,10,11,12,13].

Specifically, SRO is considered as a multiphase material consisting of a mixture of silica (SiO_2_), non-stoichiometric oxides (SiO_x_, x < 2), and elemental silicon. This material has evolved since DiMaria et al. [16] observed electroluminescence (EL) in SRO for the first time, and Leight Caham [17] obtained a visible emission of porous silicon through an electrochemical etching process. SiO_x_ materials are strong influenced by the oxygen content since it determines its optical and electrical properties such as absorption coefficient, bandgap, luminescence, refractive index, and conductivity [12,13,14].

The bonds and composition of SRO films are explained following different models; according to the random bonding model (RBM), Si and O atoms are randomly mixed in the material, and a SiO_x_ film is composed of five basic units, silicon tetrahedrons, Si-(Si_4-*n*_O*_n_*), *n* = 0, 1, 2, 3, 4, having a statistical distribution, while the mixture model (MM) suggests the existence of only Si–Si_4_ and Si–O_4_ units, neglecting the intermediate oxidation states of silicon. Finally, the interface clusters mixture model (IM) assumes the presence of both Si and SiO_2_ clusters embedded in a SiO_x_ matrix, and this is the adequate model [18].

An outstanding property of the SiOx material is the formation of silicon nanoparticles (Si-nps) embedded in the Si oxide whose atomic structure may be amorphous (a-Si), crystalline (c-Si), or a phase mixture of them. Such Si-nps result from the phase separation phenomenon of the SiO_x_ when annealing treatment is applied to the material. According to [19], the Si-nps size is determined both by the energy barrier at the interface between the Si-np and Si oxide and the annealing temperature. The degree of crystallization of the Si-nps is dependent on its size.

Thinking about optoelectronic applications, it is essential to highlight that the electrical and optical parameters of the SRO films are tunable, refractive index and conductivity can be tuned almost independently of each other over a wide range. In contrast, the refractive index is mainly determined by the [O]/[Si] ratio, and the conductivity depends on the nanostructure and the nanocrystallites of silicon [20,21].

Earlier investigations linked the high conductivity to the increased nanocrystalline content and the formation of silicon nanostructures, but in some samples, no signs of a crystalline phase are shown, which leads to the implication that such structures consist of amorphous silicon. Within inter crystallite distances of often below 1 nm, the wave functions of the electronic states generated in the crystallites can overlap, forming energy bands and allowing electron transport [20].

On the other hand, talking about the methods of obtaining the SRO, this has been manufactured by various techniques, the most common Chemical Vapor Deposition, such as Low-Pressure Chemical Vapor Deposition (LPCVD) [9,10,11,12], Hot Filament Chemical Vapor Deposition (HFCVD) [10,22,23], and Plasma Enhanced Chemical Vapor Deposition (PECVD) [4,5,6,7], in addition to sol-gel, silicone implant in SiO_2_ [24], and sputtering [25]. All these techniques are compatible with integrated circuit manufacturing technology, and they have various types of applications, such as waveguides, non-volatile memories, surge suppressors, light detection, and emission devices. The current manufacture of nanostructures is carried out mainly with materials such as porous Silicon, Silicon nanowires, Silicon Oxide Nitride (SiON), and Silicon Rich Oxide (SRO) since they are compatible with MOS technology [7,8,9,10,11,12]. Electroluminescence in MIS-type devices has been studied extensively based on different material systems, including the ZnO, perovskites, seen as [1,2]. In the present work, the HFCVD technique was used to obtain the SRO films, which are utilized to make up structures in order to study their blue electroluminescence.

## 2. Materials and Methods

The structures that were manufactured for the realization of this work were two, namely: S1 (Au/Si/SRO/ITO) and S2 (Q/Pn^+^/SRO/ITO), where Pn+ is polysilicon and Q is quartz, the structures were formed by SRO films in a simple and double layer (S-L and D-L) and with ITO contact, in order to compare the electroluminescence (EL) obtained from these devices. The methodology used to manufacture the structure S1 begins with the deposit of the SRO films by the HFCVD system. The SRO films were deposited on silicon substrates, P-type (100), 2″ diameter, with a resistivity of 1–5 Ω∙m and 300 microns thickness, previously cleaned using the standard MOS process [22,23,26] Molecular hydrogen (H_2_) fluxes (25 and 100 standard cubic centimeter per minute (sccm)) were used to grow SRO films, two samples of single-layer (S-L) (SRO_25_ and SRO_100_) and two samples of double-layer (D-L) (SRO_25/100_ and SRO_100/25_) were obtained and studied. The HFCVD system used 11 tungsten filaments energized at 74 volts with a current of 38.4 amps to reach a temperature of 2000 °C that dissociated the molecular hydrogen. The inlet of the hydrogen flow was located placed below the incandescent filaments at a distance (filament source distance, fsd) of 6 mm, obtaining thus the volatile precursors that were deposited and adsorbed on the surface of the heated substrate below the quartz sources at a distance (source substrate distance, ssd) of 8 mm [27,28]. The deposition time (dt) for the S-L SRO films was 3 min, whereas, for the D-L SRO films, it was 5 min [27,28,29,30,31,32,33]. Once the SRO films were deposited, they were annealed at 1100 °C for 60 min in an N_2_ environment. Finally, the top contact placed on the film SRO was indium tin oxide (ITO); due to these films having good transparency and conductivity, for the deposit of ITO, the spray pyrolysis method was used [31] through a nebulizer at a deposition temperature of 450 °C. The ITO solution (0.2 M) was prepared in a methanol base containing indium chloride InCl_3_ (Aldrich 99.9) and tin chloride pentahydrate SnCl_4_ 5H_2_O (Aldrich 98); the percentage of SnCl_4_ 5H_2_O was 8%. The contact used on the back of the silicon substrate was gold (Au) deposited by sputtering [32], at a vacuum pressure of 50 mTorr and a DC of 25 mA, using a gold target with a purity of 4N. On the other hand, the manufacture of structure S2 began with the deposit of the polysilicon (Pn^+^) as back contact on quartz substrates which were previously cleaned with xylene, acetone, and deionized water. It started with phosphine deposit through the LPCVD system [33], using as a precursor the SiH_4_ in an atmosphere restricted to a pressure of 1.5 torr and a flux level of 3.4 standard liters per minute (slpm), at 650 °C, for 20 min. Then phosphine doping of polysilicon was performed, and a re-diffusion time of 15 min at 1000 °C with flow levels at 1000 and 500 sccm of nitrogen and oxygen was used, respectively. It continued with the polysilicon oxidation with an oxygen flow level at 1000 sccm at 1100 °C, for 20 min. Ultimately we removed the phosphosilicate glass (PSG) glass formed on the sample surface with H_2_O: HF solution in a 7:1 ratio. After that were deposited the S-L (SRO_25_ and SRO_100_) and D-L (SRO_25/100_ and SRO_100/25_) SRO films by using the HFCVD system, exposing a strip of polysilicon (subsequent contact). Finally, the ITO was deposited as the top contact.

The thickness and refractive index of the SRO films deposited on Si substrates were characterized by the ellipsometer Fairfield Model NJ 07004-2113. All the SRO films were measurement as-grown (as-G) and with thermal annealing (T-A). The silicon excess in the composition of the SRO films was measured with an XPS and PHI ESCA-5500 using a monochromatic Al radiation source with a 1486 eV power.

For this work, the electrical characterization of the heterojunctions was performed with a Keithley source model 2400 controlled by a computer through general purpose interface bus (GPIB) using a LabVIEW^®^ software obtaining the current-voltage (I-V) curves. This equipment had the capacity to apply up to ±200 V with a resolution of 5 mV and measure from 10 pA to 105 mA [34]. To obtain EL spectra was used an optical fiber connected to the spectrometer ocean optics, which in turn connected and controlled by a computer through Spectra software.

## 3. Results and Discussion

Table 1 shows the results of the thicknesses, refractive indices (η), excess silicon (Xsi), and oxygen deficiency (X_O_) of the SRO_25_, SRO_100_, SRO_25/100_, and SRO_100/25_ films. By using Ellipsometry and XPS, characterizations to SRO films as-G and T-A were obtained. All T-A SRO films show a decrease in their thickness concerning that of as-G SRO films.

According to [8], the refractive index in the T-A S-L SRO films decreases as the oxygen in films SRO increases. This indicates an increase in the X_Si_ in the composition of the SRO films. In our T-A SRO samples refractive index decreased tending to the value of the refractive index of SiO_2_ ≈ 1.46, while the refractive index of SRO as-G tended to approach the refractive index of Si ≈ 3.42. Therefore, the results obtained contradicted what was reported in other studies; however, it was observed that for the double layer, the refractive indices and X_Si_ increased with T-A as is reported in our and other studies.

In Figure 1 the silicon and oxygen concentration profiles in XPS and the percentage of silicon and oxygen in the single-layer films of SRO_25_ and SRO_100_ are shown. It can see that the X_Si_ of the SRO films decreased from 9.9% to 5.3% and from 10.0% to 5.0% for SRO_25_ and SRO_100_ films with T-A, respectively. It can also be seen that the refractive index (η) was 2.46 and 2.04 to SRO_25_ and SRO_100_ films as-G, respectively, and with T-A, the η decreases to 1.3 and 1.02, respectively, where we had an oxygen deficiency of −9.7% and −9.5% for SRO_25_ and SRO_100_ as-G, while the oxygen deficiencies for the T-A SRO films were −6.2% and −5.0%, respectively.

To explain the behaviors that occur with respect to the silicon and oxygen contents and the refractive indices, we have that in a previous investigation [23], SRO films deposited under the same conditions and using the same HFCVD equipment as the used in this research were analyzed, in that previous research, the same behavior is observed. In this case, it is was explained that the films deposited at a lower hydrogen flux level (25 sccm) without annealing present a higher content of silicon, just as in the results of this investigation, in the HRTEM images of the quoted research, it is shown that the films tend to present agglomerations of amorphous silicon of varied sizes but in general more significant than 15 nm, in the case of higher hydrogen fluxes level (>100 sccm), the silicon content decreases, such case yields that the HRTEM images show a decrease in the size of the silicon agglomerates, when the films are thermally annealed, a diffusion of silicon and oxygen occurs where it is formed a SiO_2_–SiO_x_ matrix where silicon agglomerates with sizes smaller than 2 nm are immersed, these agglomerates show in some cases crystalline orientations, but their conformation also contains silicon in the amorphous state and oxidized terminals, this restructuring in the material causes the change in the refractive indices and silicon content.

Unusual behavior is the presence of a refractive index lower than the value of SiO_2_, which the SRO_100_ film presents after annealing, this behavior has been reported in [35], and the explanation given to this phenomenon is that SiO_x_ film has a more amorphous structure than that of the SiO_2_ film, this correlates with the fact that the SiO_x_ film is less dense and therefore has a lower refractive index [21]. This phenomenon is also discussed arguing where it is established that the crystalline regions are separated by O-rich regions, these denoting Si and O dominated areas which are clearly separated, where the decrease X_si_ is related with O-rich regions.

The different behaviors between S-L and D-L could be explained as the S-L SRO films have a thinner thickness with a deposit time of 3 min, this deposit parameter permitted to SRO films to have more amorphous silicon and X_si_ bigger, besides less X_o_ was clearly observed. Therefore, non-stoichiometric silicon oxide was much less stable, when is applied the annealing the X_si_ decreases and oxygen increases due to O-rich regions, therefore refractive index decreased. The D-L SRO films had a thicker thickness with a deposit time of 5 min, and other conditions of deposited were realized; between layer and layer deposited, there was some like annealing. Therefore, a behavior difference was obtained, and the refractive index increased in a similar manner with our other works.

All these SRO films were used to fabricate 16 structures (type MIS and MIM), of which eight structures are of type S1 (S1 = MIS) (Au/Si/SRO/ITO), and eight structures are of type S2 (S2 = MIM) (Q/Pn+/SRO/ITO), both types of structures were deposited with SL and DL both as-G and T-A. Figure 2 depicts the schematic diagram of the fabricated devices identified as S1 and S2 structures. Table 2 lists the nomenclature of the manufactured devices and their mnemonics, with which we will refer to them hereafter.

The pristine I-V curves obtained from both the as-G and T-A S1 and S2 structures with S-L and D-L are shown in Figure 3 and Figure 4, respectively. All were measured with the same voltage sweep from 0 V to 35 V after 35 V to 0 V, followed by 0 V to −35 V and closing the cycle from −35 V to 0 V, with the protection of circuit short of 100 mA. At first sight, the I-V curves corresponding to S1_as-G_ and S2_as-G_ structures exhibited current peaks with ups and downs at low voltages in both positive and negative polarizations, while those corresponding to S1_T-A_ and S2_T-A_ showed the typical characteristics of the I-V curves for MOS structures. On the other hand, it is also observed that the I-V curves illustrated higher currents for S1_25T-A_ to major voltages. The behavior of the hysteresis is shown clearly in the I-V curves of S1 and S2.

### 3.1. Curves I-V Pristine of Structures

In the pristine I-V characteristic curves of Figure 3 and Figure 4, different behaviors were identified which we will describe shortly as follows: Number 1 in these pristine I-V curves, it is observed that the S-L structures reached higher amounts of current in the first measurement as shown in Figure 3a,b than the D-L structures of Figure 3c,d. Another trend that structures presented was that films without annealing showed higher and lower peaks in current measurements at low voltages in both positive and negative polarizations. It was also observed in this behavior a sudden increase in current at a specific voltage. This phenomenon is linked with the nanostructure and its crystallinity which yields that the agglomeration of many electrons trapped in the Si-nps prevent the trapped charge’s movement and block electrical conduction [36,37,38]. Therefore, this can lead to the creation and annihilation of preferential conductive pathways generated by adjacent stable Si-nps and defects such as unstable silicon nanoclusters (Si-ncls) and others through structural changes and the possible creation of defects due to Si–O, and Si–Si [38]. Furthermore, on the return of the curve, it showed an increase in the current regarding the first measurement, where a charge trapping and state of less resistance to that of the first curve was observed. This behavior is due to the formation of conductive paths in the material; therefore, the return path was not the same, and the charge trapping was formed. This behavior is known as hysteresis [35,36,37,38]. This behavior was observed in all the I-V curves of the structures, and this effect occurred in both direct (DP) and reverse (RP) polarizations. But it is most noticeable in structures with heating treatment as well as in S-L SRO films with both polarizations, while in D-L SRO films the hysteresis in positive polarization was observed better. Number 2 in this case, the charge transport phenomenon was identified as a Coulombic Blockade, and its behavior was observed when the current increased sharply at a specific voltage, and it remained in this current as the voltage increased, this was so that since there was electrical conduction due to formation of a trapped electron configuration blocks [5], also in the resistive switching memory structures with SiO_x_ or SRO, this behavior was attributed to the presence of a point charge which induced throughout the space the appearance of a force field which broke down in the process [6,7,8,9,10,11,12,13] moving the current to a state of low resistance. Number 3 current curve was identified as a region of negative differential resistance (NDR), almost always observed after the Coulombic Block. The form it presented was a series of tiny current jumped close to each other, called resistive switching, according to Yao et al., [38,39]. This means that, for a range of values of the applied voltage, an increase in voltage caused current to decrease rather than increase and takes place when electrons traveled at the same average speed; the space charge domain no longer grew, but the electrons continued their journey, and since the electric field was not large enough to form additional domains [39,40,41], then a negative differential resistance region was created, this phenomenon was best observed in the as-G structures.

The curves of the T-A D-L S1, Figure 3c,d and the as-G S-L S2 Figure 4a,b present more significant hysteresis or charge trapping [34] than the T-A S2 Figure 4c,d and S1 as-G Figure 3a,b which means that in the T-A S1 and S-L as-G S2 samples, respond to the creation-annihilation of conductive paths due not only to the Si-ncs but also to defects in the oxide found in the heterojunctions with SRO films according to Kalnitsky et al. [41]. This negative-differential-resistance behavior in SRO/Si structures was observed through current-voltage (I-V) measurements, and the application of the electric field caused the electrical potentials to be distorted, favoring the quantum tunneling of electrons between the silicon-nanocrystal and the traps of oxide.

The graphs of current versus voltage for the S1 MIS and S2 MIM structures in Figure 5 and Figure 6 are the best of five measurements of each structure and are graphed in semi-logarithmic form. In the monolayers, we observed a current regime in the order of milliamperes when applying voltages between −25 to 25 volts, showing current variations in the as-G structures. However, the T-A structures showed a linear relationship between current and voltage until the current was maintained, and occasionally it dropped and suddenly increased, this happens again at that voltage, and current for which was possible to observe bright dots (electroluminescence) this phenomenon was observed only in the S1 MIS structures. We point out that in the S2 structures, no bright dots were observed. On the other hand, the as-G D-L structures showed abrupt increases and drops in current when increasing voltage, it was attributed to the creation and annihilation of conductive paths in the material [38,39,40,41,42,43,44,45,46]. Further, we observed the Coulombic Blockade in these T-A structures. The sweeps with forward and reverse polarization yielded a current behavior similar to that of the S-L structures but with a current regime in the order of microamperes, where at this current and with voltages greater than 30 volts it was possible to observe greater bright points, suggesting the release of charge trapping in the SRO film, generating the conductive paths at currents and voltages greater to microampere and 30 volts, respectively [41].

### 3.2. Conduction Mechanisms

The fact that the S1 and S2 structures presented similar results in forward and reverse polarization suggested that the carrier transport in this type of material was carried out through similar mechanisms [44,45] for both structures. To understand the transport mechanisms of the S-L and D-L SRO films, current density (J) measurements as a function of the electric field (E), in reverse polarization of the I-V curves plotted in Figure 5 and Figure 6 were analyzed due to these structures showed good electroluminescence. The reverse polarization occurred when the gate contact (ITO) was polarized with a negative voltage regarding substrate.

In general, four conductions mechanisms contributed to the carrier transport in these SRO films, namely: Ohmic (O), Hopping (H), Poole–Frenkel (PF), and Fowler–Nordheim (FN) [34]. The current density-electric field (J-E) analysis depended on the dielectric thickness and the electric field applied to the MOS structure.

Figure 7 and Figure 8 show the devices analyzed here in the semi-logarithmic J-E curves in reverse polarization for S1_T-A_ and S2_T-A_, structures respectively. Furthermore, in the inserts of each figure are shown, the specific J-E graphs according to the conduction mechanisms in each section of the J-E curve are highlighted with linear regions that correspond to the Ohmic (O), Hopping (H), Poole–Frenkel (PF) and Fowler–Nordheim (FN) conduction mechanisms [34,46].

As can be seen, there were several conduction mechanisms in the J-E curve of these structures. That is, for low electric fields (≤1.6 MV/cm), the carriers reached enough energy to overcome the energy barrier at the Si/SRO interface and dominate the Ohmic conduction mechanism, as is shown in the inserts of Figure 7a and Figure 8a,c. Another predominant conduction mechanism was observed in intermediate conduction regime at low electric fields, in both S-L and D-L SRO films, this is called Hopping conduction (H) [34,47,48,49,50], as seen in J-E curves in Figure 7b–d and Figure 8b,d, and it was originated by the trapped electrons jumping from one trap to another within the film SRO. Also, the energy of the trapped electrons may be less than the maximum energy of the potential barrier between the two traps; in this case, the trapped electrons may have continued traveling using the tunneling mechanism.

On the other hand, the Poole–Frenkel conduction mechanism was reported in SRO films where some electrons were found in traps and by thermal excitation were released so they could be conducted within the conduction band of the SRO films by applying an electric field through the dielectric SRO, then electrons crossed the Coulomb barrier which was reduced by the electric field and then increasing the probability of that an electron would be thermally excited from the trap becoming free to travel in the conduction band of the dielectric. The Poole–Frenkel (P-F) conduction mechanism depended strongly on the electric field, and it was independent of temperature, and the electric field was limited to low values (2 MV/cm) [34,47]; one can see this conduction mechanism in the inserts of Figure 7a and Figure 8a–d.

Additionally, the tunneling Fowler–Nordheim conduction mechanism has been proposed for the SRO films containing Si-ncs or silicon islands where electrons can tunnel by the effect of the electric field existing between the Si-ncs or silicon islands and generated by the potential barrier with a triangular shape (or another one) [34,51]. This generally occurs in not very thin dielectric films (>3.5 nm) and at high electric fields (>2 MV/cm), allowing carriers to overcome or tunnel barrier heights from one trap to another. The FN mechanism is the one that dominates [34,42,43,44,45,46,47,48,49,50,51,52].

EL emission occurs under reverse bias and is originated due to charge injection through conductive pathways and radiative recombination processes between energetic states of traps or defects [53]. It has been reported that when the electroluminescent emission is presented in the form of points, it originates from the efficiently excited emission of defects in the oxide and/or of a few Silicon nanoparticles (Si-nps) [54]. The Si-nps within the SRO films are randomly distributed so that various conductive paths are created; under this assumption, the current does not flow uniformly through the entire area of the capacitor but passes through conductive discreet paths within the oxide. An increase in the total current will result discreetly in a rise in the current density in each conductive path, which results in a more significant number of radiative recombination events and, therefore, a greater electroluminescent intensity [47]. As the current increases, more charges flow through the Si-nps and can break off some of the Si–Si bonds (creating E centers). Consequently, the conductive paths are annihilated, resulting in current drops [48].

Figure 9(a1) just shows this behavior where current increases and drops and the emission of points with different colors can be observed, it possibly indicates the intervention of different defects involved or as it is more commonly reported Si-nps of different sizes are participating [50]. In [50], it is said that electrons and holes are injected into Si-QD (quantum dots) by F-N (Fowler–Nordheim) tunneling through the SiO_x_ matrix. The existence of immersed Si-QDs leads to a decrease in the activation voltage of the F-N tunneling and creates a path for the carriers from the Si substrate to the ITO contact.

According with what it was obtained in this research, the FN conduction mechanism was presented in all structures when EL occurred, before this, the Poole–Frenkel (PF) conduction mechanism occurred at lower electric fields, which is related to electrons trapped in traps or defects which were excited towards the conduction band of the oxide. The full area EL emission obtained was due to the optimization of the injection of the carriers through the material by the cancellation of preferential conductive paths [49].

As the density of Si-nps increased, a uniform network of conductive paths became possible, allowing for uniform charge flows across the entire area. Meanwhile, as the density of Si-nps decreased, the distance between them increased, reducing the number of available paths, with a resulting set of discrete and preferential conductive pathways within the oxide. Bright spots appeared when structures were operating within a region of high conduction. These jumps and drops in luminescence were due, respectively, to the appearance and disappearance of luminescent dots on the surface of the devices. After the current drop, the EL points disappeared completely, and EL was obtained in the entire area [55].

### 3.3. Electroluminescent Structures

Figure 9(a1,b1,c1,d1) shows the reverse polarization (R.P.) I-V curves, Figure 9(a2,b2,c2,d2) shows the electroluminescent spectra, and from Figure 9(a3,b3,c3,d3), we can observe the bright dots and full electroluminescent bright area in each one of the respective photographs, belonging to the (Au/Si/SRO/ITO) S1 MIS-structures made up with S-L and D-L SRO films taking into account different deposit parameters.

From Figure 9(a1), the T-A S1_25_ MIS-structure in R.P. and X_Si_ = 5.3%, the current curve exhibited that when having −20 volts an abrupt current drop from 10 mA to 130 uA happened, at the same time brightly colored dots appeared. As the voltage varied to more negative values, the number of brightly colored dots also increased; however, due to the current drop in which it remained low, such fact provoked that the T-A S1_25_ structure did not present a uniform EL emission over the whole area. The phenomenon that we report with this structure was similar to that published in [48] when the carriers did not flow uniformly through the whole structure area, but they passed through discrete conductive paths within the SRO film, as shown in Figure 10. Consequently, the structure showed a spectrum with two outstanding peaks, one emission peak centered at around 450 nm and the other one at around 580 nm; these emission peaks remained practically at the same wavelength, but their intensities were increased as the voltages were more negative, as shown in Figure 9(a2). These EL spectra emission bands at 450 nm were associated with defects in neutral oxygen vacancies (NOVs), while the other emission of 580 nm was attributed to positively charged oxygen vacancies [10,24]. It has been reported that the EL emission peak placed in the blue band region increased its excitation voltage due to the contribution of small silicon nanoparticles (Si-nps) [44].

The EL spectra of the T-A S1_100_ MIS-structure and X_Si_ = 5.0% are shown in Figure 9(b2); we can see that when applying −15 V with a current of 63 uA, bright white dots started to appear, at once when voltage increase bright white dots were more intense, the EL spectrum intensity of the bright dots increased to the maximum emission with −25 V and 80 uA. This event was caused by holes that were attracted to the silicon surface, creating an accumulation layer, and the holes from this layer were injected toward the ITO/SRO and electrons from the ITO gate to the SRO/Si substrate interfaces. However, since the major contribution of current came from the tunneling of electrons instead of holes for MOS on p-type Si, the meeting point was closer to the SRO/Si interface, and then the recombination happened, both in the SRO film and the Si substrate surface, as reported in other works [34,44,45,46,47,48]. The latter gave rise to the EL emission spectra of the T-A S1_100_ structure showing three prominent emission peaks at around 450, 530, and 640 nm being the more intense one at 450 nm. Besides, the peaks at 530 nm of the four spectra showed a slight blue-shift. However, the peaks at 640 nm lay in this band. The two bands gave rise to intense white EL at high injection currents. The image inserted in Figure 9(b3) depicts dispersed bright white dots at −15 V with a low current of 63 uA, and such dots maintained a greater intensity when current is increased at 600 uA, which are presented as a white light spectrum, contributing to the blue, green and orange bands, such colored emissions are attributed to a competition between defects and Si-ncs, as should be expected due to the mixed material; however, as has been previously reported, the emission was mainly due to defects, especially NOV and NBOHCS ones [10,24].

Regarding the best T-A S1_25/100_ MIS-structure shown in Figure 9(c1,c2,c3), it presented a blue color full-area emission when applying −55 V at 108 uA, showing an EL emission located at the 460 nm band blue whose intensity was greater than 30,000 a.u. Figure 11 shows the progress of how the blue spectrum was emitting, starting with dots and then filling the entire area of the structure. Such an emission originated by the radiative emissions from the weak oxygen bonds (WOBs) and neutral oxygen vacancies centers (NOVs). The key factors which contributed to this emission were thermal annealing, presence of Si-related defects, D-L structure, high voltage bias with low currents, and the radiative recombination in localized states related to Si-O bonds. On the other hand, in accordance with [38,39,40,41,42,43], blue light emission was attributed to defects associated with excess silicon, which was correlated with the increase of the refractive index of 1.93 of this SRO_25/100_ films, in the same sense [47] reports that full area emission obtained is due to the optimization of carrier injection through the material by the annulation of preferential conductive paths. That is to say, [34,47] it could be related to the Si-nps density that when going to a uniform network of conductive paths, this charge uniformly flowed through the whole structure area.

Finally, the T-A S1_100/25_ MIS-structure presents an outstanding lateral emission focused mainly on its borders structure, as observed in Figure 9(d3). This phenomenon was provoked due to the establishment of conductive pathways that the Si nanocrystals generate and that allow conduction through the dielectric matrix, producing EL emission in spatial regions belonging to the thinnest layer of the film (ends). Regarding EL intensity spectra, Figure 9(d2), each broad peak was attributed to defects and Si-nps. It has also been reported that amorphous Si-nps required lower voltages but higher currents to achieve the same EL intensity as their crystalline form [56].

In Figure 9(d2), the wide-band emission of the structure is shown, spanning from 450 nm to 1000 nm. Evidently, multiple carrier recombination channels contributed to this emission spectrum. Therefore, to be sure of the emission mechanisms of the device, a deconvolution is plotted in Figure 12 to fit the peaks of the EL spectra, which according to [23] were attributed to (NBOHC) E’ ≡Si-O-O≡Si+ centers and non-bonded oxygen hole centers at the wavelengths 617 nm and 685 nm while localized luminescent centers (LLC) at the interface of nc-Si with SiO_2_ were the emission mechanisms observed in the peaks fitted for the wavelengths of 825 and 890 nm.

In this device T-A S1_100/25_, the emission was on the edge of the electrode due to the electrode having a high resistance for conduction; this did not permit the emission on the surface of the electrode as it is shown in Figure 9(d3). The conduction was easier by the electrode edge, which produced the radiative recombination and emission. The mechanism responsible for the surface-electroluminescence at the edge was related to the recombination of electron-hole pairs injected through enhanced current paths within the silicon-rich oxide film [51].

A photograph of the only T-A S2_25_ MIM-structure presented some needles-and-points of EL emission. This was reverse polarized at two different voltages, as shown in Figure 13, along with its EL response. As can be seen in Figure 13c, the central area of the structure had some bright lines and dots with increasing voltage. This can be attributed to the formation of a small number of preferential conductive pathways within the SRO film, which connected the upper electrode to the lower one. In this structure, the conduction through the active layer was not uniform but rather through discrete pathways, causing light emission to be observed only at the points corresponding to the places where conduction occurred. Additionally, there is a report [52] that oxygen-related defects rather than silicon nanocrystals are present in their SRO films with low excess silicon, the reason for which it is not possible to conform the EL emission. On the other hand, the EL spectra of the SRO film-based MIM-structure were very low intensity remaining at the wavelength between 600–700 nm as the voltage increased, shown in Figure 13b. According to [23], these are attributed to (NBOHC) E’ ≡Si-O-O≡Si+ centers and non-bonded oxygen hole centers. This behavior was similar for all values of the T-A SRO films.

Therefore, this MIM S2 structure showed no high EL activity, exhibiting only a few bright dots’ flashes, compared to the amount of more continuous dots in MIS S1 structures.

On the other hand, to obtain the emission efficiencies in the best structures, it was necessary to know the current density (J_d_) and the current density in emission (J_e_). J_d_ was obtained through the I-V curves shown in Figure 5a,c, for structures S1_25_ and S1_25/100_, respectively, while J_e_ was obtained from the I-V curves shown in Figure 9(a1,c1), for structures S1_25_ and S1_25/100_, which present electroluminescence in points and complete area, respectively. The samples S1_25_ and S1_25/100_ have the current emission densities of 317 mA/cm^2^ y 19.8 A/cm^2^, respectively. The efficiencies were obtained with the equation $\eta =\left(\frac{I_{em}}{I_{em}+I_{d}}\right)$ [57]. Therefore, at −50 V, their corresponding efficiencies were 3.2% and 19.7%, so the S_25_ and S1_25/100_ respectively. So, the S_25/100_ structure had the highest emission current density and efficiency among these structures. This result may be related to the thinnest thicknesses of the SRO films in the S1_25/100_ structure, which caused the free electrons in the SRO film to suffer less scattering during transport [57]. However, the SRO film should also not be too thin; otherwise, the film could be easily broken.

## 4. Conclusions

The S1 and S2 structures were fabricated with S-L and D-L nanometric SRO films deposited by HFCVD. It has been proved that such structures are excellent conductors and emitters. The D-L SRO structure improved the EL response compared to an S-L one. Likewise, it observed improved electrical conductivity in MIS structures with SRO films with high X_Si_ interspersed with the emitter layers, resulting in an excellent structure to be used as an electroluminescent device. In respect to the electrical and optical properties of the S1 MIS structure compared to S2 MIM structure based on S-L and D-L SRO films, it is concluded that the first structure it could be used as an efficient light source for an optoelectronic circuit, while the second was not possible to conform to the EL emission due to that oxygen-related defects rather than that silicon nanocrystals were present in their SRO films with low excess silicon. In this sense, MIS integration seems to be the best way to achieve structures for silicon optoelectronic circuits.

For all S1and S2 structures, the charge-carriers transport was dominated in different voltage domains by the Ohmic, Hopping, Poole–Frenkel and Fowler–Nordheim Conduction mechanisms, with the latter being responsible for activating the broadcast in the limit condition without the trap.

It was found the presence of a finite number of preferential conductive pathways within the SRO film which connect the upper electrode to the lower one as has been proved by the EL of the S1_25T-A_ structure whose central area exhibits bright color dots which are brighter as increasing reverse voltage. These discrete pathways define the conduction through the active layer, causing emission light only at points located at positions where the pathways exist. Thus, multiple conductive pathways will produce numerous electroluminescent spots until the full area is obtained, such as the case of the S1_25/100T-A_ structure, such structured blue light EL emission was detected whose intensity increased when increasing both reverse voltage and current. Furthermore, there is a report [47] that oxygen-related defects rather than silicon nanocrystals are present in the SRO films with low excess silicon.

## Figures and Tables

**Figure 1 nanomaterials-11-00943-f001:**
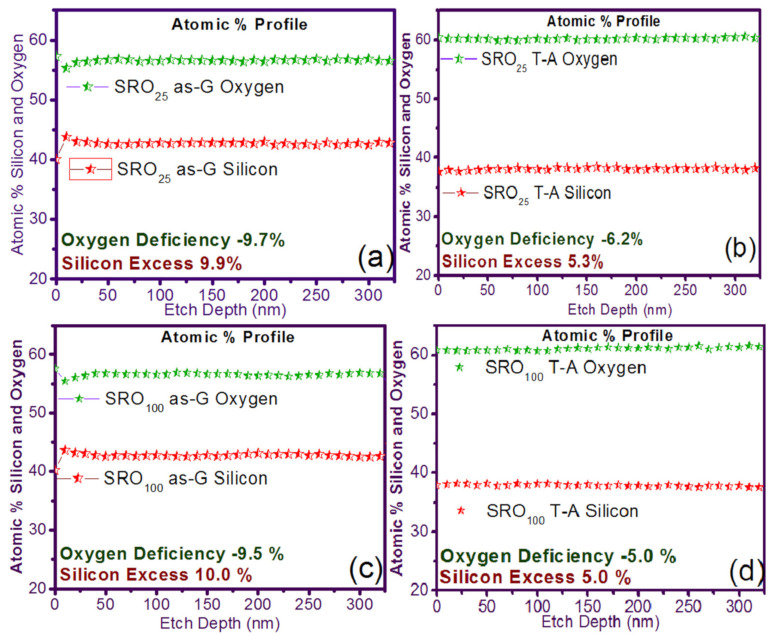
Silicon excess and oxygen deficiency for the case of simple layer (S-L) silicon-rich oxide (SRO) films (**a**) SRO_25_ as-grown (as-G), (**b**) SRO_25_ thermal annealing (T-A), (**c**) SRO_100_ as-G, and (**d**) SRO_100_ T-A.

**Figure 2 nanomaterials-11-00943-f002:**
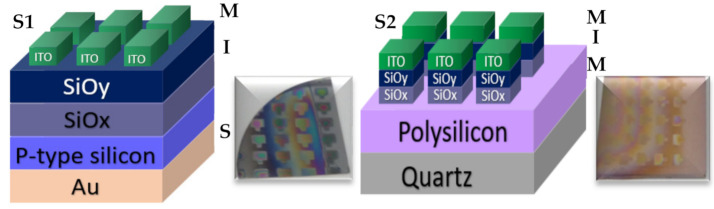
S1 (MIS) and S2 (MIM) structures fabricated D-L SRO films.

**Figure 3 nanomaterials-11-00943-f003:**
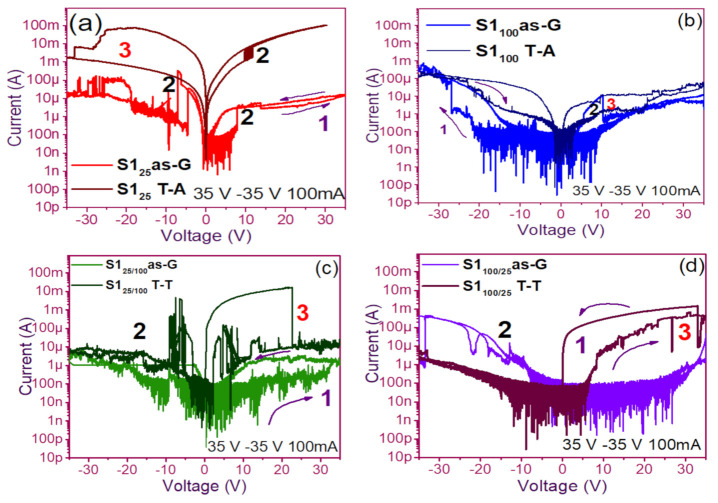
Pristine I-V curves (**a**) S1_25_ as-G and T-A, (**b**) S1_100_ as-G and T-A, (**c**) S1_25/100_ as-G and T-A, (**d**) S1_100/25_ as-G and T-A.

**Figure 4 nanomaterials-11-00943-f004:**
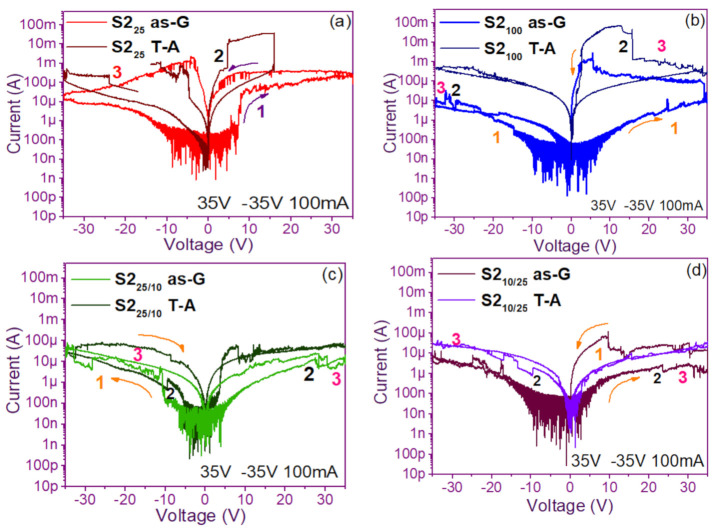
Pristine I-V curves (**a**) S2_25_ as-G and T-A, (**b**) S2_100_ as-G and T-A, (**c**) S2_25/100_ as-G and T-A, (**d**) S2_100/25_ as-G and T-A.

**Figure 5 nanomaterials-11-00943-f005:**
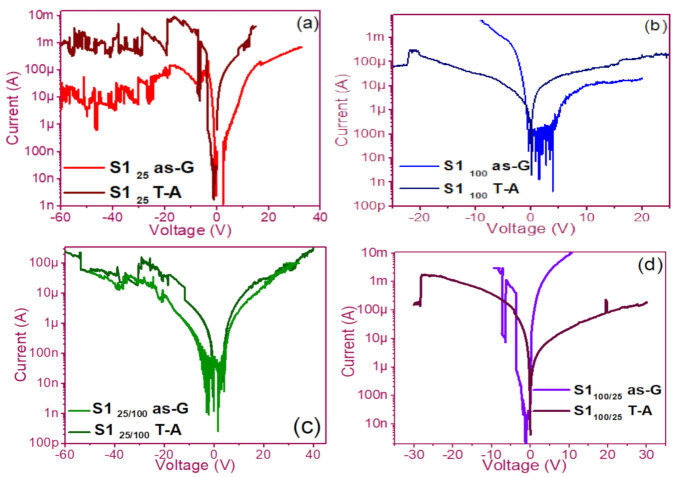
I-V curves (**a**) S1_25_ as-G and T-A, (**b**) S1_100_ as-G and T-A, (**c**) S1_25/100_ as-G and T-A, (**d**) S1_100/25_ as-G and T-A.

**Figure 6 nanomaterials-11-00943-f006:**
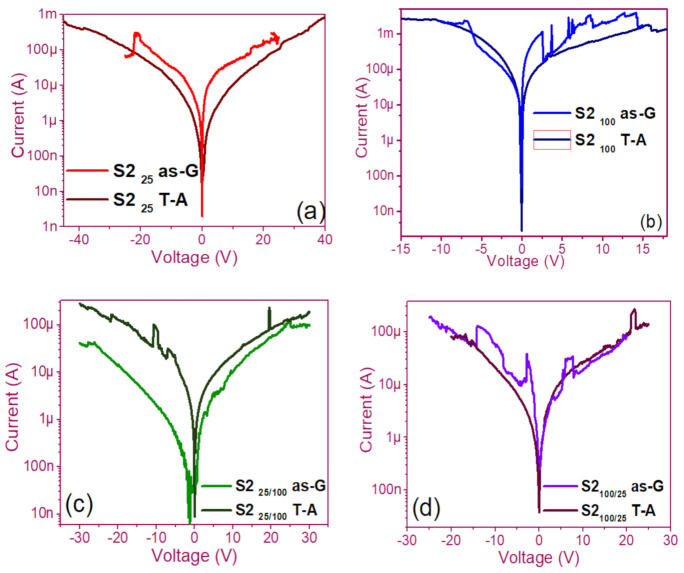
I-V curves (**a**) S2_25_ as-G and T-A, (**b**) S2_100_ as-G and T-A, (**c**) S2_25/100_ as-G and T-A, (**d**) S2_100/25_ as-G and T-A.

**Figure 7 nanomaterials-11-00943-f007:**
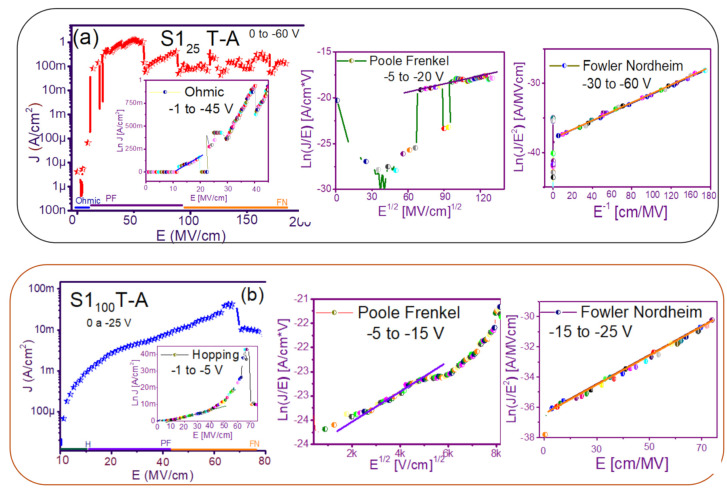
Conduction mechanisms (**a**) S1_25_ T-A, (**b**) S1_100_ T-A, (**c**) S1_25/100_ T-A, (**d**) S1_100/25_ T-A., with its corresponding J-E curve.

**Figure 8 nanomaterials-11-00943-f008:**
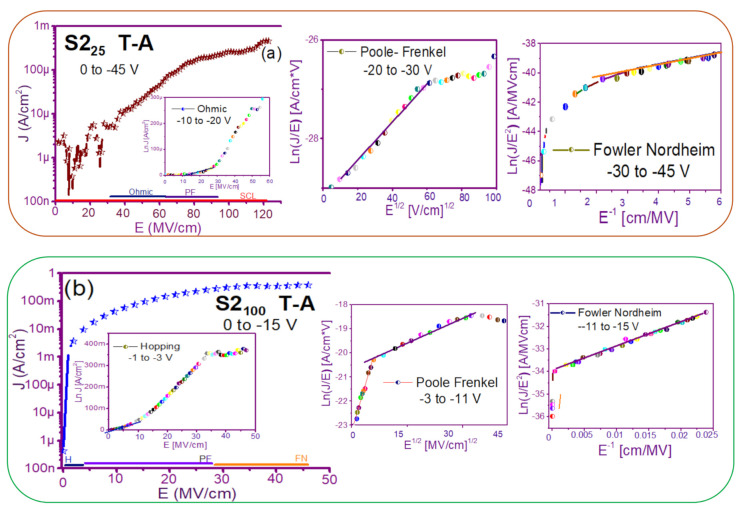
J-E curves of the different conduction mechanisms (**a**) S2_25_ T-A, (**b**) S2_100_ T-A, (**c**) S2_25/100_ T-A, (**d**) S2_100/25_ T-A.

**Figure 9 nanomaterials-11-00943-f009:**
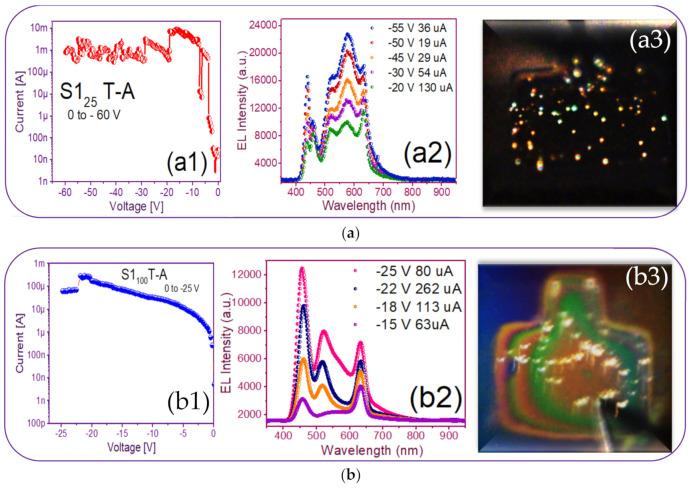
Current-voltage (I-V) and electroluminescence (EL) intensity curves and luminescent images of the (**a**) S1_25_ T-A (**b**) S1_100_ T-A (**c**) S1_25/100_ T-A (**d**) S1_100/25_ T-A.

**Figure 10 nanomaterials-11-00943-f010:**
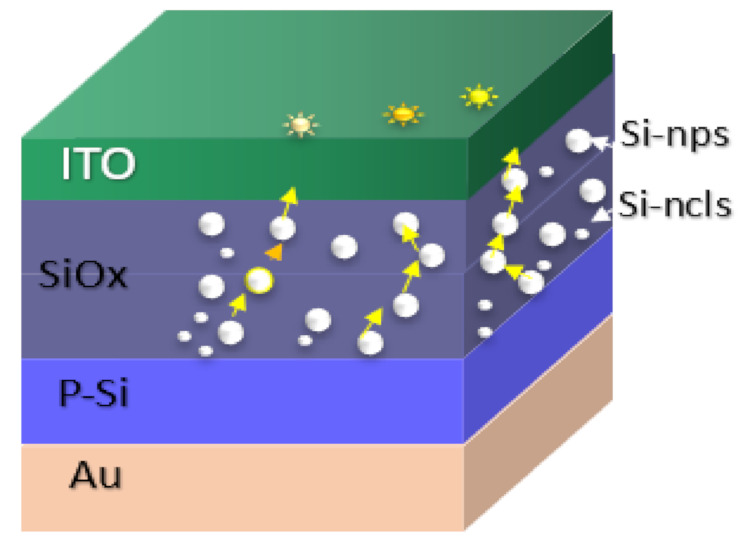
S1 structure schematic of the conductive paths within the SRO films.

**Figure 11 nanomaterials-11-00943-f011:**
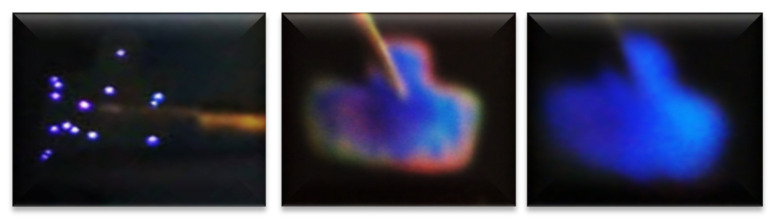
Progress of the blue emission of the complete area.

**Figure 12 nanomaterials-11-00943-f012:**
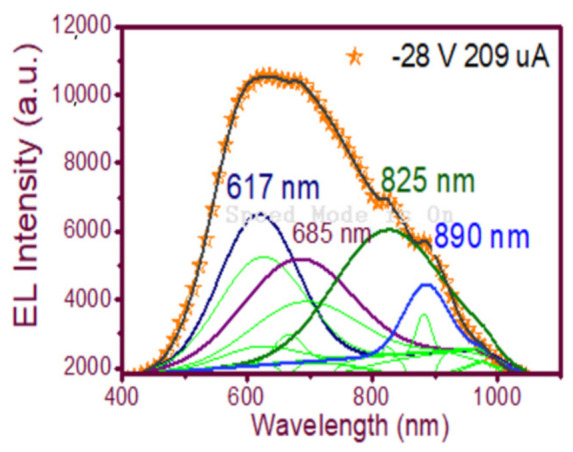
Fitting of the EL emission spectrum peaks of the wide-band EL emission spectrum of the T-A S1 _100/25_ structure.

**Figure 13 nanomaterials-11-00943-f013:**
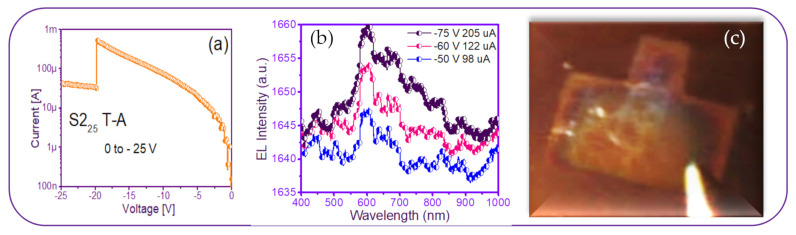
T-A S2_25_ structure (**a**) I-V curve, (**b**) EL intensity curves, and (**c**) image of the device showing a weak EL emission.

**Table 1 nanomaterials-11-00943-t001:** Thickness, refractive indices, and excess silicon of the SRO_25_, SRO_100_, SRO_25/100,_ and SRO_100/25_.

Sample	As-Grown (As-G)				Thermal Annealing (T-A)			
	Thickness nm	Refractive Indices η	X_O_%	X_Si_%	Thickness nm	Refractive Indices η	X_O_%	X_Si_%
Si/SRO_25_	322.9	2.46 ± 0.03	−9.7	9.9	296.3	1.30 ± 0.04	−6.2	5.3
Q/Pn^+^/SRO_25_	351.8				319.6			
Si/SRO_100_	319.6	2.04 ± 0.35	−9.5	10.0	283.5	1.02 ± 0.08	−5.0	5.0
Q/Pn^+^/SRO_100_	339.0				306.6			
Si/SRO_25/100_	592.3	1.46 ± 0.06			583.3	1.93 ± 0.31		
Q/QPn^+^/SRO_25/100_	654.3				611.7			
Si/SRO_100/25_	578.9	1.51 ± 0.002			560.5	2.05 ± 0.65		
Q/QPn^+^SRO_100/25_	668.2				599.3			

**Table 2 nanomaterials-11-00943-t002:** Metal Insulator Semiconductor and Metal Insulator Metal structures with S-L and double layer (D-L) SRO films both as-G and T-A.

Structure 1 (S1)	Mnemonics	Structure 2 (S2)	Mnemonics
MIS		MIM	
Au/Si/SRO_25as-G_/ITO	S1_25as-G_	Q/Pn+/SRO_25as-G_/ITO	S2_25as-G_
Au/Si/SRO_100as-G_/ITO	S1_100as-G_	Q/Pn+/SRO_100as-G_/ITO	S2_100as-G_
Au/Si/SRO_25/100as-G_/ITO	S1_25/100as-G_	Q/Pn+/SRO_25/100as-G_/ITO	S2_25/100as-G_
Au/Si/SRO_100/25as-G_/ITO	S1_100/25as-G_	Q/Pn+/SRO_100/25as-G_/ITO	S2_100/25as-G_
Au/Si/SRO_25T-A_/ITO	S1_25T-A_	Q/Pn+/SRO_10T-A_/AZO	S2_25T-A_
Au/Si/SRO_100T-A_/ITO	S1_100T-A_	Q/Pn+/SRO_25T-A_/AZO	S2_100T-A_
Au/Si/SRO_25/100T-A_/ITO	S1_25/100T-A_	Q/Pn+/SRO_25/10T-A_/AZO	S2_25/100T-A_
Au/Si/SRO_100/25T-A_/ITO	S1_100/25T-A_	Q/Pn+/SRO_10/25T-A_/AZO	S2_100/25T-A_

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
