# Peer review of "Blue Electroluminescence in SRO-HFCVD Films"

_nanomaterials, 2021, doi:10.3390/nano11040943_

Round 1
Reviewer 1 Report
This report presents the results of a study on the electroluminescence of MIS and MIM type structures. These structures are fabricated by single and double layer hot filament chemical vapor deposition of silicon-rich oxide films and gold and indium tin oxide contacts on silicon and quartz substrates. The thickness, refractive index, and silicon excess of the SRO films are investigated, and the carrier transport in the SRO films. The conduction mechanisms that contribute to the carrier transport in the SRO films were shown. These conduction mechanisms are related to the electroluminescence spectra obtained from MIS and MIM-like structures with SRO films. It will be potentially possible to fabricate it could be used as a light source for an optoelectronic circuit and should be in front of the rear device. However, the reviewer asks to revise and elaborate on the following some points;
- The drawings in Figure 2 are unclear to read. Please translate characters in S1 and S2 to English. The reviewer requests that they should be improved.
- The captions in Figures 1, 3, 4, 5, and 6 and photos in Figures 9 and 10 should be clarified. Their captions including axis and chart are very small and vaguely.
- The reviewer was able to confirm that the emission patterns of the MIS and MIM structures differed depending on the structure, but the explanation of the mechanism for the difference in the planar or spot-like emission is unclear. The mechanism of conductivity can be inferred from the IV characteristics, but a detailed explanation of the correlation with the emission spectrum is needed.
For above reasons, the reviewer cannot recommend the present manuscript for the publication of this journal, however, if the answers to the above two points can be verified, the reviewer supports the acceptance of this manuscript.
Author Response
Reviewer 1
- The drawings in Figure 2 are unclear to read. Please translate characters in S1 and S2 to English. The reviewer requests that they should be improved.
Response 1: Figure 2 was improved. In addition, the labels of the structures in Figure 2 were translated.
- The captions in Figures 1, 3, 4, 5, and 6 and photos in Figures 9 and 10 should be clarified. Their captions, including axis and chart, are very small and vaguely.
Response 2: The captions in Figures 1, 3, 4, 5, and 6 have been clarified, and their size was increased. The photographs of Figures 9 and 10 (now 13) were improved and clarified.
- The reviewer was able to confirm that the emission patterns of the MIS and MIM structures differed depending on the structure, but the explanation of the mechanism for the difference in the planar or spot-like emission is unclear. The mechanism of conductivity can be inferred from the IV characteristics, but a detailed explanation of the correlation with the emission spectrum is needed.
Response 3: In order to answer this point, four paragraphs were added in which it is written in greater detail how the electroluminescent emission is obtained in MIS and MIM structures. These are shown below and are marked in red in the article in lines 342 to 377 of pages 7 and 8.
EL emission occurs under reverse bias and is originated due to charge injection through conductive pathways and radiative recombination processes between energetic states of traps or defect [56]. It has been reported that when the electroluminescent emission is presented in the form of points, it originates from the efficiently excited emission of defects in the oxide and/or of a few Silicon nanoparticles (Si-nps) [57]. The Si-nps within the SRO films are randomly distributed so that various conductive paths are created; under this assumption, the current does not flow uniformly through the entire area of the capacitor but passes through conductive paths discreet within the oxide. An increase in the total current will result discreetly in a rise in the current density in each conductive path, which results in a more significant number of radiative recombination events and, therefore, a greater electroluminescent intensity [58]. As the current increases, more charges flow through the Si-nps and can break off some of the Si-Si bonds (creating E centers). Consequently, the conductive paths are annihilated, resulting in current drops [59].
Figure 9 (a1) just shows this behavior where current increases and drops and the emission of points with different colors can be observed, it possibly indicates the intervention of different defects involved or as it is more commonly reported Si-nps of different sizes are participating [60]. In [60], it is said that electrons and holes are injected into Si-QD (quantum dots) by F-N (Fowler-Nordheim) tunneling through the SiOx matrix. The existence of immersed Si-QDs leads to a decrease in the activation voltage of the F-N tunneling and creates a path for the carriers from the Si substrate to the ITO contact.
According to what was obtained in this research, the FN conduction mechanism is presented in all structures when occurring EL, before this, the PF (Poole-Frenkel) conduction mechanism occurs at lower electric fields, which is related to electrons trapped in traps or defects which are excited towards the conduction band of the oxide. The full area EL emission obtained was due to the optimization of the injection of the carriers through the material by the cancellation of preferential conductive paths [61].
As the density of Si-nps increases, a uniform network of conductive paths becomes possible, allowing for uniform charge flows across the entire area. Meanwhile, as the density of Si-nps decreases, the distance between them increases, reducing the number of available paths, with a resulting set of discrete and preferential conductive paths within the oxide. Bright spots appear when structures are operating within a region of high conduction. These jumps and drops in luminescence are due, respectively, to the appearance and disappearance of luminescent dots on the surface of the devices. After the current drop, the EL points disappear completely, and EL is obtained in the entire area [62].
[56] Leyva, K. M., Aceves-Mijares, M., Yu, Z., Flores, F., Morales-Sánchez, A., & Alcántara, S. (2010). Visible electroluminescence on FTO/thin SRO/n-Si structures. Materials Science and Engineering: B, 174(1-3), 141-144.
[57] Valenta, J., Lalic, N., & Linnros, J. (2001). Electroluminescence microscopy and spectroscopy of silicon nanocrystals in thin SiO2 layers. Optical Materials, 17(1-2), 45-50.
[58] Morales-Sánchez, A., Barreto, J., Domínguez, C., Aceves, M., Leyva, K. M., Luna-López, J. A., ... & Pedraza, J. (2010). Topographic analysis of silicon nanoparticles-based electroluminescent devices. Materials Science and Engineering: B, 174(1-3), 123-126.
[59] Morales-Sánchez, A., Barreto, J., Domínguez, C., Aceves-Mijares, M., Perálvarez, M., Garrido, B., & Luna-López, J. A. (2010). DC and AC electroluminescence in silicon nanoparticles embedded in silicon-rich oxide films. Nanotechnology, 21(8), 085710.
[60] Cheng, C. H., Lien, Y. C., Wu, C. L., & Lin, G. R. (2013). Mutlicolor electroluminescent Si quantum dots embedded in SiO x thin film MOSLED with 2.4% external quantum efficiency. Optics Express, 21(1), 391-403.
[61] Fernández, A. G., Mijares, M. A., Sánchez, A. M., & Leyva, K. M. (2010). Intense whole area electroluminescence from low pressure chemical vapor deposition-silicon-rich oxide based light emitting capacitors. Journal of Applied Physics, 108(4), 043105.
[62] Morales-Sánchez, A., Monfil-Leyva, K., González, A. A., Aceves-Mijares, M., Carrillo, J., Luna-López, J. A., ... & Flores-Gracia, F. J. (2011). Strong blue and red luminescence in silicon nanoparticles based light emitting capacitors. Applied Physics Letters, 99(17), 171102.

Reviewer 2 Report
The research focuses on the studying of electroluminescence in MIS and MIM-type structures based on SRO thin films obtained by HW CVD method. The improvement of EL response for double layer SRO structures compared to using single-layer SRO structures. However, some important explanations (in particular related to the structure and optical properties of the SRO) should be given.
- Please provide a more detailed introduction to SRO thin films (Introduction part), structure and optoelectronic material properties. While expanding this part, the authors may consider works reported on the same topic, such as https://doi.org/10.1002/pssa.201533022 https://doi.org/10.1016/j.jnoncrysol.2019.05.015 https://doi.org/10.1063/1.4953566.
- Some of the results obtained require clarification. You associate the decline in XSi with the reorganization of the structural net in the SRO. At the same time, you are talking about the possible formation of Si nanocrystallites (nanoclusters) during annealing, which, on the contrary, implies an increase in XSi. The main mechanism of phase separation is the diffusion of oxygen atoms from regions with a lower concentration to the regions with a higher concentration, while the diffusion of silicon is vice versa [https://doi.org/10.1002/pssa.201900513]. How can you comment on this contradiction? Could the decrease in XSi be related to the oxidation of SRO during annealing in a nitrogen atmosphere containing oxygen contamination? If you have confirmation of the formation of Si nanocrystallites (nanoclusters) after annealing?
- I am very confused by the values of the refractive index of SRO100 after annealing (1.02 ± 0.08), which is much less than the value of SiO2, but at the same time you claim that an excess of Si is still present in these films. How is this possible?
- How can you explain that for single-layer samples SRO25 and SRO100 there is a tendency to a decrease in the refractive index after annealing, but for a two-layer structure SRO25/100 the refractive index increases?
Author Response
Reviewer 2
- Please provide a more detailed introduction to SRO thin films (Introduction part), structure, and optoelectronic material properties. While expanding this part, the authors may consider works reported on the same topic, such as https://doi.org/10.1002/pssa.201533022 https://doi.org/10.1016/j.jnoncrysol.2019.05.015 https://doi.org/10.1063/1.4953566.
Response 1: With reference to this suggestion, six paragraphs were added to increase the introduction according to the reviewer comments. These are shown below and are marked in red in the article in lines 36 to 37 and others in lines 45 to 54, and others in lines 65 to 92, and on pages 1 and 2.
Although in the field of optoelectronics there has been substantial research regarding electroluminescent MIM structures [1,2], the use of silicon-based materials, structures, and devices is a crucial advantage for optoelectronic applications due to the high-performance and low-cost manufacturing techniques that are used [3-6], besides that silicon is the second most abundant chemical element in the earth. Silicon-based structures focused on optoelectronics include Si nanocrystals, Si nanowires, Ge alloys, and other methods to achieve the required carrier confinement to increase the efficiency of radiative recombination [7,8], and it is essential to mention that significant advances are also being realized with silicon oxide materials containing optically active rare-earth impurities [7,8].
Some of the most exciting applications of Si-nanostructures include optical emitters for integrated optical circuits, memory logic circuits; massively parallel optical interconnects and cross-connections for integrated circuit chips; light-wave components; high power matrix and discrete emitters; nano cell optoelectronic matrices to detect biological and chemical agents [9-16].
The semiconductor nanostructures based on silicon that absorb and emit light, such as porous silicon (PS), silicon nanowires, silicon oxide nitride (SiON), and silicon-rich oxide (SRO), present a great opportunity of being applied in electro photonics and they are compatible with MOS technology [7-13].
Specifically, SRO is considered as a multiphase material consisting of a mixture of silica (SiO2), non-stoichiometric oxides (SiOx, x <2), and elemental silicon.
This material has evolved since DiMaria et al. [17] observed electroluminescence (EL) in SRO for the first time, and Leight Caham [18] obtained a visible emission of porous silicon through an electrochemical etching process.
SiOx materials are strong influenced by the oxygen content since it determines its optical and electrical properties such as absorption coefficient, bandgap, luminescence, refractive index, and conductivity [12-14].
The bonds and composition of SRO films are explained following different models; according to the random bonding model (RBM), Si and O atoms are randomly mixed in the material, and a SiOx film is composed of five basic units, silicon tetrahedrons, Si-(Si4- nOn), n=0, 1, 2, 3, 4, having a statistical distribution, while the mixture model (MM) sug-gests the existence of only Si-Si4 and Si-O4 units, neglecting the intermediate oxidation states of silicon. Finally, the interface clusters mixture model (IM) assumes the presence of both Si and SiO2 clusters embedded in a SiOx matrix, and this is the adequate model [19].
An outstanding property of the SiOx material is the formation of silicon nanoparticles (Si-nps) embedded in the Si oxide whose atomic structure may be amorphous (a-Si), crystalline (c-Si), or a phase mixture of them. Such Si-nps result from the phase separation phenomenon of the SiOx when annealing treatment is applied to the material. According to [20], the Si-nps size is determined both by the energy barrier at the interface between the Si-np and Si oxide and the annealing temperature. The degree of crystallization of the Si-nps is dependent on its size.
Thinking about optoelectronic applications, it is essential to highlight that the electrical and optical parameters of the SRO films are tunable, refractive index and conductivity can be tuned almost independently of each other over a wide range. In contrast, the refractive index is mainly determined by the [O]/[Si] ratio, and the conductivity depends on the nanostructure and the nanocrystallites of silicon [21,22].
Earlier investigations linked the high conductivity to the increased nanocrystalline content and the formation of silicon nanostructures, but in some samples, no signs of a crystalline phase are shown, which leads to the implication that such structures consist of amorphous silicon. Within inter crystallite distances of often below 1 nm, the wave function of the electronic states generated in the crystallites can overlap, forming energy bands and allowing electron transport [21].
[19] Zamchiy, A. O., Baranov, E. A., Merkulova, I. E., Khmel, S. Y., & Maximovskiy, E. A. (2019). Determination of the oxygen content in amorphous SiOx thin films. Journal of Non-Crystalline Solids, 518, 43-50.
[20] A. Sarikov (2020). Crystallization Behavior of Amorphous Si Nanoinclusions Embedded in Silicon Oxide Matrix. Phys. Status Solidi A, 217, 1900513. https://doi.org/10.1002/pssa.201900513).
[21] Klingsporn, M., Kirner, S., Villringer, C., Abou-Ras, D., Costina, I., Lehmann, M., & Stannowski, B. (2016). Resolving the nanostructure of plasma-enhanced chemical vapor deposited nanocrystalline SiOx layers for application in solar cells—Journal of Applied Physics, 119(22), 223104.
[22] Smirnov, V., Lambertz, A., Moll, S., Bär, M., Starr, D. E., Wilks, R. G., ... & Finger, F. (2016). Doped microcrystalline silicon oxide alloys for silicon‐based photovoltaics: Optoelectronic properties, chemical composition, and structure studied by advanced characterization techniques. Physica status solidi (a), 213(7), 1814-1820.
- Some of the results obtained require clarification. You associate the decline in XSi with the reorganization of the structural net in the SRO. At the same time, you are talking about the possible formation of Si nanocrystallites (nanoclusters) during annealing, which, on the contrary, implies an increase in XSi. The main mechanism of phase separation is the diffusion of oxygen atoms from regions with a lower concentration to the regions with a higher concentration, while the diffusion of silicon is vice versa [https://doi.org/10.1002/pssa.201900513]. How can you comment on this contradiction? Could the decrease in XSi be related to the oxidation of SRO during annealing in a nitrogen atmosphere containing oxygen contamination? If you have confirmation of the formation of Si nanocrystallites (nanoclusters) after annealing?
Response 2: With reference to this response, one paragraph was added. This is shown below and is marked in red in the article in lines 180 to 195 on page 4.
To explain the behaviors that occur with respect to the silicon and oxygen contents and the refractive indices, we have that in a previous investigation [37], SRO films deposited under the same conditions and using the same HFCVD equipment as the used in this research were analyzed, in that previous research, the same behavior is observed. In this case, it is was explained that the films deposited at a lower hydrogen flux (25 sccm) without annealing present a higher content of silicon, just as in the results of this investigation, in the HRTEM images of the quoted research, it is shown that the films tend to present agglomerations of amorphous silicon of varied sizes but in general more significant than 15 nm, in the case of higher hydrogen fluxes (100 sccm), the silicon content decreases, such case yields that the HRTEM images show a decrease in the size of the silicon agglomerates, when the films are thermally annealed, a diffusion of silicon and oxygen occurs where it is formed a SiO2-SiOx matrix where silicon agglomerates with sizes smaller than 2 nm are immersed, these agglomerates show in some cases crystalline orientation, but their conformation also contains silicon in the amorphous state and oxidized terminals, this restructuring in the material are what cause the change in the refractive in-dices and silicon content.
[37] Hernández Simón, Z. J., Luna López, J. A., de la Luz, A. D. H., Pérez García, S. A., Benítez Lara, A., García Salgado, G., ... & Martínez Hernández, H. P. (2020). Spectroscopic Properties of Si-nc in SiOx Films Using HFCVD. Nanomaterials, 10(7), 1415.
- I am very confused by the values of the refractive index of SRO100 after annealing (1.02 ± 0.08), which is much less than the value of SiO2, but at the same time, you claim that an excess of Si is still present in these films. How is this possible?
Response 3: With reference to this response, one paragraph was added. This is shown below and is marked in red in the article in lines 197 to 204 on page 4.
Unusual behavior is the presence of a refractive index lower than the value of SiO2, which the SRO100 film presents after annealing, this behavior has been reported in [38], and the explanation given to this phenomenon is that SiOx film has a more amorphous structure than that of the SiO2 film, this correlates with the fact that the SiOx film is less dense and therefore has a lower refractive index, in [22]. This phenomenon is also discussed where it is established that crystalline regions are separated by O-rich regions, these denoting Si and O dominated areas which are clearly separated, where the decrease Xsi is related with O-rich regions.
[38] Ma, H. P., Yang, J. H., Yang, J. G., Zhu, L. Y., Huang, W., Yuan, G. J., ... & Lu, H. L. (2019). Systematic study of the SiOx film with different stoichiometry by plasma-enhanced atomic layer deposition and its application in SiOx/SiO2 superlattice. Nanomaterials, 9(1), 55.
[22] Smirnov, V., Lambertz, A., Moll, S., Bär, M., Starr, D. E., Wilks, R. G., ... & Finger, F. (2016). Doped microcrystalline silicon oxide alloys for silicon‐based photovoltaics: Optoelectronic properties, chemical composition, and structure studied by advanced characterization techniques. Physica status solidi (a), 213(7), 1814-1820.
- How can you explain that for single-layer samples SRO25 and SRO100, there is a tendency to a decrease in the refractive index after annealing, but for a two-layer structure, SRO25/100, the refractive index increases?
Response 4: With reference to this response, one paragraph was added. This is shown below and is marked in red in the article in lines 205 to 213 on page 5.
The different behaviors between S-L and D-L could be explained as SRO films with simple-layer have a thinner thickness with a deposit time of 3 minutes, these deposited parameters permitted to SRO films to have more amorphous silicon and Xsi bigger, is clearly observed less Xo. Therefore, non-stoichiometric silicon oxide is lesser stable, when is applied the annealing the Xsi decreases and oxygen increases due to O-rich regions, therefore refractive index decreases. The D-L SRO films have a thicker thickness with a deposit time of 5 minutes, and other conditions of deposited were realized; between layer and layer deposited, there is some like annealing. Therefore, a behavior difference is obtained, and the refractive index increases in a similar manner with our other works

Reviewer 3 Report
- In Figure 9d2, the device emits a broadband emission covering from 450 nm to 1000 nm. Obviously, the emission spectra is contributed by multiple carrier recombination channels. So, to make certain the emission mechanisms of the device, peak fitting of the EL spectra should be conducted and related explanations on the fitted peaks are needed.
- I do not know why the device emits light at the edge of the electrode, not on the surface of the electrode (Figure 9d3).
- In Figure 7, the insets are too small to distinguish the words.
- Electroluminescence in MIS-type device has been studied extensively based on different material systems, including the ZnO, perovskites, seen as ACS Appl. Mater. Interfaces, 2018, 10, 32289; Optics Letters, 2016, 41, 5608.
- So many language errors and format errors in this article, such as Figure 9 and Figure 10b, “Voltage should be abbreviated as “V”, not “v”. Besides, the quality of figures need to be improved.
Author Response
Reviewer 3
- In Figure 9d2, the device emits a broadband emission covering from 450 nm to 1000 nm. Obviously, the emission spectra are contributed by multiple carrier recombination channels. So, to make certain the emission mechanisms of the device, peak fitting of the EL spectra should be conducted, and related explanations on the fitted peaks are needed.
Response 1: With reference to this response, one paragraph and on Figure (12) were added. This is shown below and is marked in red in the article in lines 447 to 454 on pages 10 and 18.
In Figure 9 (d2), the wide-band emission of the device is shown, spanning from 450 nm to 1000 nm. Evidently, multiple carrier recombination channels contribute to this emission spectrum. Therefore, to be sure of the emission mechanisms of the device, a deconvolution is plotted in Figure 12 to fit the peaks of the EL spectra, which according to [23] are attributed to (NBOHC) E' ≡Si-O-O≡Si+ centers and non-bonded oxygen hole centers at the wavelengths 617 nm and 685 nm while localized luminescent centers (CLI) at the interface of nc-Si with SiO2 are the emission mechanisms observed in the peaks fitted for the wavelengths of 825 and 890 nm.
- I do not know why the device emits light at the edge of the electrode, not on the surface of the electrode (Figure 9d3).
Response 2: With reference to this response, one paragraph was added. This is shown below and are marked in red in the article in the lines 455 to 460 in the pages 10.
In this device S1 100/25 T-A, the emission is on the edge of the electrode due to the electrode have a high resistance for conduction; this did not permit the emission on the surface of the electrode. The conduction is easier by the edge of the electrode, which produces the radiative recombination and emission. The mechanism responsible for the sur-face-electroluminescence at the edge has been related to the recombination of electron-hole pairs injected through enhanced current paths within the silicon-rich oxide film [65].
[65] Morales-Sánchez, A., Domínguez, C., Barreto, J., Aceves-Mijares, M., Licea-Jiménez, L., Luna-López, J. A., & Carrillo, J. (2013). Floating substrate luminescence from silicon rich oxide metal-oxide-semiconductor devices. Thin solid films, 531, 442-445.
- In Figure 7, the insets are too small to distinguish the words.
Response 3: Figure 7 was improved; the insets have been expanded, and the words can now be distinguished
- Electroluminescence in MIS-type devices has been studied extensively based on different material systems, including the ZnO, perovskites, seen as ACS Appl. Mater. Interfaces, 2018, 10, 32289; Optics Letters, 2016, 41, 5608.
Response 4:
As a suggestion to the reviewer, the references mentioned have been added. This is shown below and is marked in red in the article on lines 104 to 106 on pages 3.
Electroluminescence in MIS-type devices has been studied extensively based on different material systems, including the ZnO, perovskites, seen as [25,26].
- [25] Shi, Z., Lei, L., Li, Y., Zhang, F., Ma, Z., Li, X., ... & Du, G. (2018). Hole-injection layer-free perovskite light-emitting diodes. ACS applied materials & interfaces, 10(38), 32289-32297.
- [26] Shi, Z., Li, Y., Zhang, Y., Wu, D., Xu, T., Zhang, B., ... & Du, G. (2016). Electrically pumped ultraviolet lasing in polygonal hollow microresonators: an investigation on optical cavity effect. Optics Letters, 41(23), 5608-5611.
- So many language errors and format errors in this article, such as Figure 9 and Figure 10b, “Voltage should be abbreviated as “V”, not “v”. Besides, the quality of figures needs to be improved.
Response 5: The v for V was corrected, and the quality of the figures was improved.

Reviewer 4 Report
The authors present a nice study on the electroluminescence (EL) from silicon-rich-oxide films. The manuscript is well organized. The results of this work are helpful for the understanding of EL property from SRO materials, which, as introduced in the manuscript, can find their advantages over other EL materials in certain important application scenarios. I suggest that this work can be accepted for publication on Nanomaterials. The following questions are presented for the authors' consideration: 1) Is it possible to conduct temperature dependent IV measurement to give more information on analyzing conduction mechanisms? 2) How is the emission efficiency for these devices? 3) Please check the spell of some terminology again for consistency. E.g, in Fig. 4 and Fig. 5, the spell of current and voltage is different.
Author Response
Reviewer 4
The following questions are presented for the authors' consideration:
1) Is it possible to conduct temperature-dependent IV measurement to give more information on analyzing conduction mechanisms?
Response 1:
At this moment, it is not possible due to that there is no Access to the laboratory.
2) How is the emission efficiency for these devices?
Response 2: With reference to this response, one paragraph was added. These are shown below and are marked in red in the article in lines 180 to 195 on page 4.
In order to obtain the emission efficiencies in the best structures, it is necessary to know the current density (Jd) and the current density in emission (Je). Jd was obtained through the I-V curves shown in Figures 5 (a) and 5 (c), for structures S125 and S125/100, respectively, while Je was obtained from the I-V curves shown in Figures 9 (a1) and 9 (c1), for structures S125 and S125/100, which present electroluminescence in points and complete area, respectively. The samples S125 and S125/100 have the current emission densities of 317 mA/cm2 y 19.8 A/cm2, respectively. The efficiencies were obtained with the equation h=(I_em/(I_em+I_d )) [68]. Therefore, at -50 V, their corresponding efficiencies were 3.2 % and 19.7 %, so the S25 and S125/100 respectively. So, the S25/100 structure has the highest emission current density and efficiency among these structures. This result may be related to the thinnest thicknesses of the SRO films in the S125/100 structure, which causes that the free electrons in the SRO film to suffer less scattering during transport [68]. However, the SRO film should also not be too thin; otherwise, the film can be easily broken.
[68] Negishi Nobuyasu, Chuman Takashi, Iwasaki Shingo, Yoshikawa Takamasas, Ito Hiroshi and Ogasawara Kiyohide. (1997). High-Efficiency Electron-Emission in Pt/Siox/Si/Al Structure. The Japan Society of Applied Physics, 36, L 939-L941
3) Please check the spell of some terminology again for consistency. E.g, in Fig. 4 and Fig. 5, the spell of current and voltage is different.
Response 3:
Spelling errors in all figures have been corrected.

Round 2
Reviewer 1 Report
Thank you for replying against the reviewer's comments. The reviewer recommends to publish the present form.
Reviewer 2 Report
Thank you for your comprehensive answers. The manuscript can be published in its present form.